# Evaluation of an Isothermal Amplification HPV Assay on Self-Collected Vaginal Samples as Compared to Clinician-Collected Cervical Samples

**DOI:** 10.3390/diagnostics13213297

**Published:** 2023-10-24

**Authors:** Aaron H. Y. Chan, Siew-Fei Ngu, Lesley S. K. Lau, Obe K. L. Tsun, Hextan Y. S. Ngan, Annie N. Y. Cheung, Karen K. L. Chan

**Affiliations:** 1Department of Obstetrics and Gynaecology, School of Clinical Medicine, Queen Mary Hospital, The University of Hong Kong, Hong Kong; chy947@hku.hk (A.H.Y.C.); lsk382@hku.hk (L.S.K.L.); hysngan@hku.hk (H.Y.S.N.); kklchan@hku.hk (K.K.L.C.); 2Department of Pathology, Queen Mary Hospital, The University of Hong Kong, Hong Kong; obetsun@pathology.hku.hk (O.K.L.T.); anycheun@pathology.hku.hk (A.N.Y.C.)

**Keywords:** cervical cancer screening, human papilloma virus (HPV), self-sampling, Sentis HPV, isothermal amplification, BD Onclarity, Hong Kong

## Abstract

This study aimed to evaluate the concordance of HPV results between the Sentis^TM^ HPV assay (Sentis) (BGI Group, Shenzhen, China), an isothermal amplification-based HPV assay, on self-collected and clinician-collected samples and the agreement of Sentis on self-collected samples with the BD Onclarity^TM^ HPV assay (Onclarity) (Becton, Dickinson, and Company, Franklin Lakes, New Jersey, USA), a PCR-based HPV assay, on clinician-collected samples. This was a prospective study of 104 women attending the colposcopy clinic for abnormal smears. After informed consent, participants self-collected vaginal samples before having clinician-collected cervical samples. Self-collected samples underwent HPV testing with Sentis (Self-Sentis HPV) and clinician-collected samples were tested with Sentis (Clinician-Sentis HPV) and Onclarity (Clinician-Onclarity), which was used as a reference standard. The concordance was assessed using Cohen’s kappa. The prevalence of HPV and the acceptability of self-sampling were also evaluated. The concordance rate between Self-Sentis HPV and Clinician-Sentis HPV was 89.8% with a kappa of 0.769. The concordance rate between Self-Sentis HPV and Clinician-Onclarity was 84.4% with a kappa of 0.643. The prevalence of HPV was 26.0% on Clinician-Onclarity, 29.3% on Clinician-Sentis HPV, and 35.6% on Self-Sentis HPV. Overall, 65% of participants would undergo self-sampling again. This was attributed to mainly not feeling embarrassed (68%) and being convenient (58%). Our study showed a substantial agreement between Self-Sentis HPV with Clinician-Sentis HPV and Clinician-Onclarity. Self-sampling was also shown to be a generally well-accepted method of screening.

## 1. Introduction

Cervical cancer ranks fourth amongst the most common female cancers in the world with around 604,000 women diagnosed with the disease and caused 342,000 deaths in 2020 [1]. In Hong Kong, cervical cancer was the seventh most common cancer and also the eighth leading cause of cancer deaths amongst females in 2020 [2]. According to a report by the Hong Kong Department of Health in 2020, only 45.8% of women aged between 25 and 64 years were ever screened for cervical cancer [3]. In 2022, only 21.3% of the local population in the same age group were registered under the cervical cancer screening programme [4]. In contrast, statistics from some developed countries show that up to 70% of eligible individuals aged between 25 to 64 years were screened adequately [5]. The cause for this low uptake is multifactorial and includes the efficacy of outreach programmes and socio-cultural barriers, which may include embarrassment, fear and discomfort of gynaecological examination, inconvenience, lack of time, and associated costs [6,7].

The World Health Organisation (WHO) has recently developed a global strategy for cervical cancer elimination to be achieved by 2030. One action of this strategy is to utilise a high-performance test in at least 70% of women with at least two screenings at 35 and 45 years of age [8]. The advantages of using human papillomavirus (HPV) testing as the primary cervical cancer screening method versus cytology alone or co-testing are well established worldwide [9,10,11]. Likewise, in Hong Kong, the use of HPV testing has been found to be more sensitive than traditional cytology in detecting high-grade cervical intraepithelial lesions (CIN2+) [12]. As we inevitably move towards an era of using HPV testing as the primary screening method for cervical cancer, it has also come to the attention of cervical screening programmes around the world that we may be able to increase the uptake of cervical cancer screening via self-sampling for HPV [13,14,15]. HPV self-sampling has the potential to break some socio-cultural barriers associated with cervical cancer screening by eliminating the need for visits to the clinic. By offering an option for women to perform self-sampling in the comfort of their own homes, we can provide an alternative to clinician-based screening for non-attenders and potentially reduce the incidence of cervical cancer and associated mortality.

Various devices are commercially available and have been used for self-sampling. Recent meta-analyses have also shown similar sensitivity in samples collected by the women themselves and clinicians when validated polymerase chain reaction (PCR) based HPV tests were used [16,17]. For any screening test, it is important to be reliable, accurate, and easy to perform as well as having a low cost to the healthcare system. Existing PCR-based HPV assays can be costly and complex to perform. Hence, newer technologies have been developed to provide similar reliability and accuracy while reducing the costs and complexity of HPV testing. Isothermal amplification of nucleic acids, a simple procedure that accumulates nucleic acid sequences rapidly and efficiently at a constant temperature has been developed as an alternative to PCR. HPV assays utilising isothermal amplification techniques have recently been compared with traditional PCR-based HPV assays and show promising data to suggest their use in cervical cancer screening [18,19]. Before utilising these new assays for self-sampling in local screening programmes, it is important to evaluate their accuracy on self-samples against clinician-collected samples as sample preparation and analysis may differ significantly. The Sentis^TM^ HPV assay (Sentis) is an isothermal amplification real-time fluorescent HPV detection assay. The test is based on assays that have been shown to have similar sensitivity to PCR-based HPV tests between self-sampled and clinician-sampled results for the detection of CIN2+ [19]. It has also been shown to have an almost perfect agreement between clinician-collected samples when compared with PCR-based assays [18]. This study aimed to evaluate the concordance of HPV results between Sentis on self-collected and clinician-collected samples as well as self-collected samples using Sentis with BD Onclarity^TM^ HPV assay (Onclarity), a reference standard PCR-based test, on clinician-collected samples.

## 2. Materials and Methods

This study was conducted in accordance with the ethical principles of the latest Declaration of Helsinki and was approved by the local institution’s ethics review board (The University of Hong Kong and Hospital Authority Hong Kong West Cluster Institutional Review Board). The primary outcome of this study was to evaluate the concordance of HPV results using Sentis between self-collected vaginal samples and clinician-collected cervical samples and also the agreement of self-collected vaginal samples using Sentis with Onclarity using clinician-collected cervical samples. Secondary outcomes included the concordance between Sentis and Onclarity on clinician-collected cervical samples, the prevalence of HPV infections in self-collected and clinician-collected samples as well as the attitudes of women towards HPV self-sampling.

Participants were recruited from the colposcopy clinic at Queen Mary Hospital, a university-affiliated hospital with an annual new colposcopy referral of around 650 women. Women attending the clinic for abnormal smears or for a follow-up for previous abnormal smears were eligible. Inclusion criteria included patients aged between 30 and 65 years with a history of sexual activity. Exclusion criteria included a history of hysterectomy or being pregnant or menstruating at the time of recruitment.

An information sheet about the study was gone through with the women and written informed consent was obtained. Participants were first shown an instructional video on how to collect self-samples. A private room was provided for self-sampling. They were then given a self-collection toolkit that included a set of diagrammed instructions for self-sampling, a sealed long sterile brush (Cepillo Endocervical/Cervical Brush/Cyto-Brush, Ningbo HLS Medical Products Co. Ltd., China), and a DNA sample storage card (solid transport media) (BGI Biotechnology (Wuhan) Co. Ltd., Wuhan, China) which was used to collect specimens from the brush (Figure 1). They were instructed to insert the brush into the upper vagina, rotate 3 to 5 times, and then remove the brush from the vagina.

The brush was then brushed back and forth over the sample collection patch on the DNA sample storage card and left to dry in a safe place. The dried storage card was then sealed in a provided Ziploc bag. After self-sampling, the patient attended a consultation with a gynaecologist who collected a cervical smear using a Cervex-Brush (Rovers Medical Devices, Oss, The Netherlands) placed in a liquid-based medium (ThinPrep Pap Test, Hologic, Marlborough, MA, USA) and using a second Cervex-Brush to collect a cervical sample applied over a DNA sample storage card. Participants also completed an acceptability questionnaire which included sociodemographic data and attitudes towards their experience with self-sampling. The acceptability of self-sampling was evaluated using a five-point Likert scale that included a range of subjective qualities such as convenience, embarrassment, confidence, discomfort, and overall experience.

Clinician-collected samples underwent standard cytological examination and HPV testing using Sentis [CE-marked; BGI Group, China] and Onclarity, which was used as a reference standard. The self-sampled specimens were processed with Sentis only. Of the three HPV tests performed, participants were only informed of their HPV results from the clinician-collected samples analysed using Onclarity. HPV results from Sentis were performed for research purposes and were not used for direct patient care.

Any abnormal results were managed according to the Hong Kong College of Obstetricians and Gynaecologists (HKCOG) cervical cancer prevention and screening guidelines (2016) [20]. Clinicians also performed colposcopy examination and cervical biopsy when indicated according to the HKCOG guidelines.

Data were collected and analysed using Statistical Package for the Social Sciences version 26 (SPSS Inc., Chicago, IL, USA). Women with incomplete data (inadequate specimen, failure of any HPV test) were excluded from the analysis. The concordance rate of HPV between self-collected samples and clinician-collected samples was calculated by absolute agreement and Cohen’s kappa statistics, which was interpreted as no agreement (≤0), slight (0.01–0.20), fair (0.21–0.40), moderate (0.41–0.60), substantial (0.61–0.80) and almost perfect agreement (0.81–1.00).

### 2.1. Sentis^TM^ HPV Assay (Sentis)

Sentis is an isothermal amplification real-time fluorescent HPV detection assay, developed by BGI Group, Shenzhen, China. The assay is based on the AmpFire HPV assay and was supplied by Atila Biosystems (Sunnydale, California, USA). Specific primers and probes are used to amplify regions of viral genomic DNA including E6/E7 regions. The assay detects 14 high-risk (hr) HPV subtypes (individual detection of 16, 18, and pooled detection of 31, 33, 35, 39, 45, 51, 52, 56, 58, 59, 66, and 68). The self-sampling kits were provided by Sunrise Diagnostic Centre (SDC) Limited and Sentis was performed by BGI Health (HK) Company Limited in Hong Kong according to the manufacturer’s instructions. The accuracy and reproducibility of AmpFire HPV assay for primary cervical cancer screening have previously been validated [18,19].

### 2.2. BD Onclarity^TM^ HPV Assay (Onclarity)

Onclarity (Becton, Dickinson, and Company, Franklin Lakes, New Jersey, USA) is a United States Food and Drug Administration (FDA) approved HPV assay and is used as the reference standard. It is a fully automated real-time PCR assay that detects 14 hrHPV subtypes. The test specifically identifies 6 hrHPV for types 16, 18, 31, 45, 51, and 52 while concurrently detecting 8 other hrHPV types (33/58, 35/39/68 and 56/59/66). Along with the cytological and histological samples, Onclarity was performed by the Department of Pathology, at the University of Hong Kong according to standard protocols and the manufacturer’s instructions.

## 3. Results

A total of 104 eligible women were recruited between April and July 2022. The median age of participants was 45 years (range 30–64) and 81% were Chinese. Of these, 101 women had self-collected samples processed with Sentis (Self-Sentis HPV), 100 had clinician-collected samples processed for cytology and Onclarity (Clinician-Onclarity), and 99 had clinician-collected samples processed with Sentis (Clinician-Sentis HPV). Between 3–5 women in each group did not have their samples processed due to missing samples.

The concordance of Self-Sentis HPV and Clinician-Sentis HPV samples was 89.8% with a positive agreement of 93.1%, a negative agreement of 88.4%, and a kappa of 0.769, indicating substantial agreement. The concordance of Self-Sentis HPV and Clinician-Onclarity samples was 84.4% with a positive agreement of 88.5%, a negative agreement of 82.9%, and a kappa of 0.643, also indicating substantial agreement. The concordance of Clinician-Sentis HPV and Clinician-Onclarity samples was 95.8% with a positive agreement of 96.1%, a negative agreement of 95.6%, and a kappa of 0.897, indicating almost perfect agreement (Table 1).

For the reference standard test using Onclarity on clinician-collected samples, the prevalence of hrHPV amongst our study population was 26.0%. For Clinician-Sentis HPV and Self-Sentis HPV samples, the prevalence of hrHPV was 29.3% and 35.6%, respectively (Table 2).

A total of 3 patients had a high-grade squamous intraepithelial lesion (HSIL) diagnosed on cervical biopsy, and all had positive hrHPV results on Self-Sentis HPV and Clinician-Sentis HPV. However, one out of the three patients did not have hrHPV detected on Clinician-Onclarity. For patients with a histological low-grade squamous intraepithelial lesion (LSIL) on cervical biopsy, 12 (63.2%), 8 (44.4%), and 6 (31.6%) patients had hrHPV detected on Self-Sentis HPV, Clinician-Sentis HPV, and Clinician-Onclarity, respectively. For patients who had normal histology on cervical biopsy, 6 (50.0%), 7 (63.6%), and 7 (58.3%) had hrHPV detected on Self-Sentis HPV, Clinician-Sentis HPV, and Clinician-Onclarity, respectively. Cytology results were available for 95 patients (5 had unsatisfactory results or the test was not completed) (Table 3).

There were 100 acceptability questionnaires received. Four patients did not return the questionnaires. Of the 100 respondents, 68% were not embarrassed or not embarrassed at all, 58% found it convenient or very convenient, and 46% found it an easy or very easy process. However, only 34% were confident or very confident about collecting the sample correctly, and 31% had an overall good or very good experience with self-sampling (Table 4). Furthermore, 65% of women would undergo self-sampling again, mostly because they found the procedure simple (75%) and quick (48%). However, 33% of participants would not consider self-sampling again due to a lack of confidence in taking self-sampling accurately (67%) and found the procedure painful (30%) (Table 3). The most preferred method for cervical cancer screening among the participants was the physician conducting a speculum examination for Pap smear (45%). (Table 4)

## 4. Discussion

To justify the use of HPV self-sampling in a local screening program, it is important to show adequate accuracy of such screening methods before considering its application. It is also paramount to validate a specific HPV assay according to sample collection methods in order to obtain reliable and reproducible results [21]. PCR-based assays are currently the gold standard for HPV detection for both research and commercial testing, attributable to their high sensitivity, specificity, and reliability [22,23,24]. Due to an inherent need for easily accessible self-sampling tests to increase the reach of cervical cancer screening, research has recently been focused on comparing self- and clinician-collected samples using PCR-based assays. A recent meta-analysis showed that the sensitivity and specificity between self-collected and clinician-collected samples were similar when using PCR-based tests [17]. Arbyn et al. reported a concordance of 88.7% with a kappa of 0.72, a positive agreement of 84.6%, and a negative agreement of 91.7% [25]. Furthermore, the concordance of HPV testing among clinician- and self-collected samples was reported to range from 78.2% to 96.9% in a recent systemic review [26]. These are promising results for the promotion of self-sampling; however, PCR-based assays are notoriously complex and have high operational costs. This is mainly due to the need for sophisticated laboratories providing thermal cyclers or complex thermal equipment to run the assays. Given the potential high costs and complexity of PCR-based assays, it was imperative for public health to develop tests of similar accuracy and reproducibility with lower costs and complexity. Isothermal amplification has been gaining attention amongst researchers in providing amplification with high sensitivity and specificity, allowing for high-throughput analysis at a lower cost compared with PCR-based techniques [27,28,29]. It allows for deoxyribose nucleic acid (DNA) amplification at a constant temperature and does not require complex thermal equipment [30].

The AmpFire HPV assay developed by Atila Biosystems, USA, is an isothermal amplification assay designed to detect hrHPV. The use of the assay has shown promising data in providing similar accuracy in self-collected and clinician-collected samples when compared with PCR-based assays. Zhang et al. showed that the AmpFire HPV assay had similar sensitivity when compared to PCR-based assays in a large cohort [19]. Desai et al. studied the concordance between a similar test using isothermal amplification and a PCR-based assay for clinician-collected samples and found an unweighted kappa of 0.89 (almost perfect agreement) [18]. Due to the simplicity of running the assay without specialized equipment, it has provided a method for rapid, high-volume, and low-cost HPV detection for cervical cancer screening [31]. Furthermore, it does not require a specialized medium for transportation and dry self-collected specimens can be stored for up to two weeks [32]. Other isothermal amplification assays for HPV detection are also available [33,34]. Together with AmpFire, they all show promise in providing a rapid, accurate, and cost-effective method for HPV detection. We anticipate that large-scale comparative studies will be performed as the technology matures.

In our study, the concordance rate between clinician-collected samples for Sentis and Onclarity was 95.8% with a kappa of 0.89, which is the same as that found by Zhang et al. More importantly for the purpose of our study, we found a substantial agreement between Self-Sentis HPV with Clinician-Sentis HPV (concordance of 89.8% and a kappa of 0.769) as well as with Clinician-Onclarity (concordance of 84.4% and a kappa of 0.643). To our knowledge, this is the first concordance study between self-collected and clinician-collected samples using an AmpFire-based assay for hrHPV detection. This substantial agreement provides preliminary support for the use of isothermal amplification techniques for HPV testing in cervical screening and has the potential to reach non-attenders by use of self-sampling.

As we move away from the COVID-19 pandemic, the possibilities in terms of telemedicine have been broadened and accepted in many parts of the world. Self-sampling kits for rapid antigen testing and telemedicine consultations were commonplace, reducing the workload on healthcare professionals and allowing patients to be managed remotely [35]. The dawn of primary HPV screening has also allowed more accessible means of cervical cancer screening. The lessons learnt from the COVID-19 pandemic can be applied to using self-sampling for cervical cancer screening to increase screening rates and reach those who would otherwise be non-attenders. The effectiveness of a send-to-all approach was previously shown to be more effective than an opt-in strategy for HPV self-sampling [36]. This can increase screening participation amongst high-risk non-attenders and may be a strategy worth considering in local screening programs [37]. The use of tests using isothermal amplification techniques can lower the costs and can be of benefit if such a strategy were to be applied.

The prevalence of hrHPV in our study population ranged from 26 to 36%, depending on the method of sample collection and test performed. This number is much higher than that observed in other local studies, which ranged from 4.5% to 12.9% [38,39]. This may be explained by the fact that our study population was patients already being followed up or being referred to our colposcopy clinic for an abnormal smear. Hence these women would be expected to have a higher prevalence of HPV infection. It is interesting to note that Self-Sentis HPV returned the highest percentage of hrHPV whereas the clinician-collected reference standard Onclarity test returned the lowest. This is likely contributed by the different collection tools and methods, HPV assays used, and laboratory where the tests were performed. When considering a screening test, it is important not to overdiagnose, which may cause unnecessary investigations and potential harm. On the other hand, it is also important not to miss a diagnosis causing delay in treatment and affecting prognosis.

In the current study, three patients who underwent colposcopy and cervical biopsy had HSIL on histology. Both self- and clinician-collected samples tested with Sentis^TM^ HPV assay detected hrHPV associated with HSIL in all three patients whereas clinician-collected samples tested with Onclarity only detected hrHPV in two out of three patients. HPV 16 was detected in four patients, and the results were consistent regardless of the collection method or HPV assay used. Another patient had a double positive result with HPV 16 and another hrHPV type detected on Self-Sentis HPV, but not detected on Clinician-Sentis HPV or Clinician-Onclarity. According to our findings, it seems that the HPV result of both self- and clinician-collected samples analysed using Sentis is at least as accurate as the reference standard of clinician-collected samples tested with Onclarity. Furthermore, HPV testing on both self- and clinician-collected samples seems to be better in detecting CIN2+ than the traditional cytology as one of the three HSIL patients returned a negative cytology result. Although limited by the small number of patients, our findings are consistent with the available evidence supporting the use of HPV self-sampling using isothermal amplification techniques in cervical cancer screening [19].

Our study found that the majority of patients (65%) would undergo self-sampling again. Women found it an easy and convenient method of cervical cancer screening which minimized embarrassment. However, the majority of participants who would not consider self-sampling again cited a lack of confidence in taking self-sampling accurately. Therefore, measures that may increase confidence in performing self-collection, such as the appropriate instructions and design of the collection kits, are important. Our participants were women who were already attending colposcopy services for abnormal cervical smears. These women are attenders and their attitudes to self-sampling may differ from those who have never been screened or are long-term non-attenders. Ngu et al. had previously shown that the acceptability of self-sampling in the local never- or under-screened population was similar in terms of previous smear experience, parity, and educational background. The uptake rate of self-sampling was up to 62% [39]. Given these results and the results from the current study, the general acceptability of self-sampling and its use can be utilised to improve our local uptake in those who are never or under screened, and also provide a means for the continuity of cervical cancer screening for attendees who may potentially default follow-ups.

One of the limitations of this study is its small sample size and short recruitment time. However, given the available data on isothermal amplification techniques for HPV testing, our data suggests a similar almost perfect agreement between clinician-collected samples for Sentis and PCR-based Onclarity. More significantly, we found a substantial agreement between Self-Sentis HPV and Clinician-Sentis as well as Self-Sentis HPV and Clinician-Onclarity. This study has provided preliminary evidence for the use of self-sampling using isothermal amplification techniques for HPV testing in our local population. Another limitation is that the participants were recruited from the colposcopy clinic rather than in the primary screening setting. Nonetheless, given that the aim of our study was to compare the concordance of HPV results between self- and clinician-collected samples with different HPV assays, we would require an adequate number of HPV-positive women to achieve this.

## 5. Conclusions

Our study has shown a substantial agreement between self- and clinician-collected samples for HPV screening using isothermal amplification techniques, which has also been shown to be comparable to the reference standard. Self-sampling was also shown to be a generally well-accepted method of screening. In a population that is prone to socio-cultural barriers, it is important to provide an alternative to traditional clinician-led screening methods. This study has provided preliminary evidence to support self-sampling using self-sampling kits and isothermal amplification assays in the local cervical screening program.

## Figures and Tables

**Figure 1 diagnostics-13-03297-f001:**
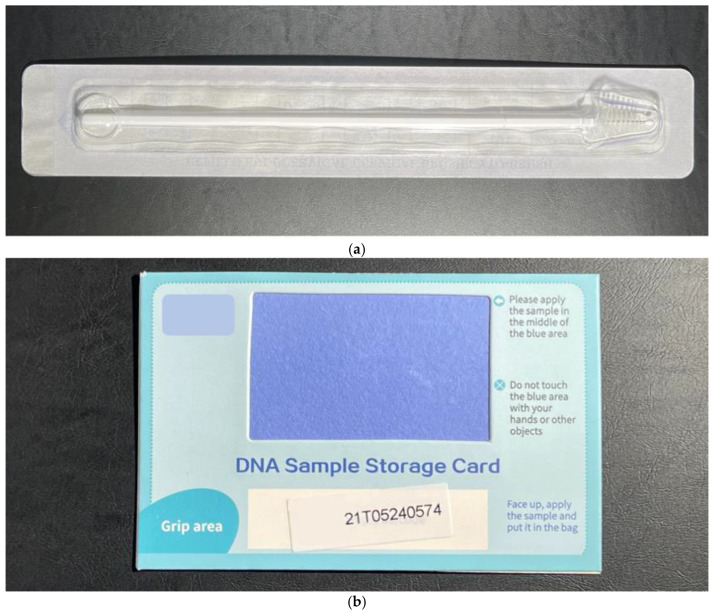
HPV Self-collection toolkit containing a sealed long sterile brush (**a**) and a DNA sample storage card (**b**).

**Table 1 diagnostics-13-03297-t001:** Concordance of self-collected Sentis^TM^ HPV samples (Self-Sentis HPV), clinician-collected Sentis^TM^ HPV samples (Clinician-Sentis HPV), and clinician-collected BD Onclarity (Clinician-Onclarity) samples.

Test	Comparator Test (CT)	HPV Test Result	CT Negative (*n*)	CT Positive (*n*)	Concordance (%)	Positive Agreement (%)	Negative Agreement (%)	Kappa
Self-Sentis HPV	Clinician-Sentis HPV	Negative	61	2	89.8%	93.1%	88.4%	0.769
		Positive	8	27				
Self-Sentis HPV	Clinician-Onclarity	Negative	58	3	84.4%	88.5%	82.9%	0.643
		Positive	12	23				
Clinician-Sentis HPV	Clinician-Onclarity	Negative	66	1	95.8%	96.1%	95.6%	0.897
		Positive	3	25				

**Table 2 diagnostics-13-03297-t002:** Prevalence of high-risk human papillomavirus infections in clinician and self-collected samples.

HPV Test	Clinician-Onclarity *n* (%)	Clinician-Sentis HPV *n* (%)	Self-Sentis HPV *n* (%)
hrHPV			
Positive	26 (26.0)	29 (29.3)	36 (35.6)
Negative	71 (71.0)	70 (70.7)	64 (63.4)
Inconclusive	3 (3.0)	0 (0)	1 (1.0)
hrHPV subtypes			
16	4 (4.0)	4 (4.0)	5 (5.0) ^#^
18	0 (0)	0 (0)	0 (0)
Others *	22 (22.0)	25 (25.3)	31 (30.6)

hrHPV, high-risk human papillomavirus. * Others refers to hrHPV subtypes 31, 33, 35, 39, 45, 51, 52, 56, 58, 59, 66, and 68. ^#^ One patient had a double positive with another hrHPV subtype.

**Table 3 diagnostics-13-03297-t003:** Cytology and histology results compared with clinician-collected samples for BD Onclarity^TM^ (Clinician-Onclarity) and Sentis^TM^ HPV (Clinician-Sentis HPV) as well as self-collected samples for Sentis ^TM^ HPV (Self-Sentis HPV).

		High-Risk HPV Results			
Clinician-Onclarity		Negative (*n* = 71)	Positive (*n* = 26)	Inconclusive (*n* = 3)	Total (*n* = 100)
Cytology	Negative	51	10	1	62
	ASCUS	15	11	1	27
	LSIL	0	4	0	4
	ASC-H	1	0	0	1
	HSIL	0	1	0	1
	Unsatisfactory	4	0	1	5
Histology	No Biopsy	53	11	2	66
	Negative	5	7	0	12
	LSIL	12	6	1	19
	HSIL	1	2	0	3
Clinician-Sentis HPV		Negative (*n* = 70)	Positive (*n* = 29)	Inconclusive (*n* = 0)	Total (*n* = 99)
Cytology	Negative	52	10	0	62
	ASCUS	13	13	0	26
	LSIL	0	4	0	4
	ASC-H	0	1	0	1
	HSIL	1	0	0	1
	Unsatisfactory	3	1	0	4
	Not completed	1	0	0	1
Histology	No Biopsy	56	11	0	67
	Negative	4	7	0	11
	LSIL	10	8	0	18
	HSIL	0	3	0	3
Self-Sentis HPV		Negative (*n* = 64)	Positive (*n* = 36)	Inconclusive (*n* = 1)	Total (*n* = 101)
Cytology	Negative	50	12	0	62
	ASCUS	9	17	1	27
	LSIL	0	4	0	4
	ASC-H	0	1	0	1
	HSIL	1	0	0	1
	Unsatisfactory	3	2	0	5
	Not completed	1	0	0	1
Histology	No Biopsy	50	15	1	66
	Negative	6	6	0	12
	LSIL	7	12	0	19
	HSIL	0	3	0	3

ASCUS, atypical squamous cells of undetermined significance; LSIL, low-grade squamous intraepithelial lesions; ASC-H, atypical squamous cells, cannot exclude high-grade intraepithelial lesions; HSIL, high-grade squamous intraepithelial lesions.

**Table 4 diagnostics-13-03297-t004:** Acceptability questionnaire—willingness to undergo HPV self-sampling again and the most preferred method for cervical cancer screening.

Attitudes of Women towards Self-Sampling	*N* = 100 (%)
Overall good or very good experience	31%
Easy or very easy	46%
Convenient or very convenient	58%
Not embarrassed or not embarrassed at all	68%
No discomfort or no discomfort at all	33%
Confident or very confident	34%
Would you be willing to do the HPV self-sampling test again?	*N* (%)
Yes	65 (65.0)
Reason *	
The test is simple to do	49 (75.4)
The test is quick	31 (47.7)
Taking a sample with a swab was not painful	20 (30.8)
I feel less embarrassed taking my own sample	19 (29.2)
I feel more comfortable taking own sample	15 (23.1)
I am confident that I can take my own sample accurately	10 (15.4)
No	33 (33.0)
Reason *	
I am not confident in taking my own sample accurately	22 (66.7)
Taking my own sample with the swab was painful	10 (30.3)
I am afraid I might hurt myself	9 (27.3)
I prefer a healthcare professional to collect the sample	8 (24.2)
The test is not easy	6 (18.2)
I am not comfortable taking my own sample	4 (12.1)
Unanswered	2 (2.0)
Which method do you prefer the most for cervical screening?	*N* (%)
Pap smear—physician conducting a speculum examination	45 (45.0)
Self-collecting vaginal swab for HPV testing	22 (22.0)
Physician-collected vaginal swab for HPV testing	21 (21.0)
No preference	12 (12.0)

* More than one answer was allowed.

## Data Availability

Not applicable.

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
