# Peer review of "Evaluation of an Isothermal Amplification HPV Assay on Self-Collected Vaginal Samples as Compared to Clinician-Collected Cervical Samples"

_diagnostics, 2023, doi:10.3390/diagnostics13213297_

Round 1
Reviewer 1 Report
Comments:
1. The title of this study is “evaluation of an isothermal amplification HPV Assay on…”, but isothermal amplification was not mentioned in the manuscript. It is suggested to add some background information about isothermal amplification and discuss about it in the section of discussion section.
2. A total of 104 eligible women were recruited between April and July 2022. The sample quantity is relatively small and the sample sampling time span is relatively short.
3. The authors stated that a substantial agreement between self and clinician-collected samples for HPV screening using isothermal amplification techniques, which has also been shown to be comparable to the reference standard. However, the concordance rate between Self-Sentis HPV and Clinician-Sentis HPV was 89.8% with a kappa of 0.769. The concordance rate between Self- Sentis HPV and Clinician-Onclarity was 84.4% with kappa of 0.643. Whether the consistency is high enough indicates the accuracy of Self-sampling?
4. What is the minimum detection line of method Self-Sentis HPV?
5. Except for the method mentioned in this study, are there any other methods used to detect HPV? And what are the advantages of the Self-Sentis HPV compared with them?
6. The format of table 3 is not standard. Please redo the drawing.
7. The formatting of the references should keep consistent.
Need for moderate editing of English language
Reviewer 2 Report
The authors evaluated the performance of an Isothermal amplification based HPV assay, SentisTM HPV, for the potential screening of cervical cancer. The assay was combined with a self-sampling kit which allowed an individual to independently collect the specimen needed for HPV test. This approach could potentially improve cervical cancer screening and reduce the embarrassment associated with specimen collection during clinic visitation. Overall, the manuscript was well written but could still benefit from some language editing.
The authors indicated that the Sentis HPV assay was less expensive and complex compared to PCR based methods. However, the authors need to provide evidence to support these claims. Furthermore, several Isothermal amplification methods exist in literature and commercially, the authors should describe the Isothermal method used in this study. These should be further discussed along with the benefits and limitations of their study.
Overall, the manuscript was well written but could still benefit from language editing
Round 2
Reviewer 2 Report
No further comments
OK, Minor edit will improve the manuscript